# High-Pressure Technologies for the Recovery of Bioactive Molecules from Agro-Industrial Waste

Junyang Li [1], Margherita Pettinato [1], Roberta Campardelli [1,*], Iolanda De Marco [2,*] and Patrizia Perego [1]

1 Department of Civil, Chemical and Environmental Engineering (DICCA), University of Genoa, Via Opera Pia 15, 16145 Genova, GE, Italy; junyang.li@edu.unige.it (J.L.); margherita.pettinato@edu.unige.it (M.P.); p.perego@unige.it (P.P.)
2 Department of Industrial Engineering, University of Salerno, Via Giovanni Paolo II, 132, 84084 Fisciano, SA, Italy
* Correspondence: roberta.campardelli@unige.it (R.C.); idemarco@unisa.it (I.D.M.)

**Abstract:** Large amounts of food waste are produced each year. These residues require appropriate management to reduce their environmental impact and, at the same time, economic loss. However, this waste is still rich in compounds (e.g., colorants, antioxidants, polyphenols, fatty acids, vitamins, and proteins) that can find potential applications in food, pharmaceutical, and cosmetic industries. Conventional extraction techniques suffer some drawbacks when applied to the exploitation of food residues, including large amounts of polluting solvents, increased time of extraction, possible degradation of the active molecules during extraction, low yields, and reduced extraction selectivity. For these reasons, advanced extraction techniques have emerged in order to obtain efficient residue exploitation using more sustainable processes. In particular, performing extraction under high-pressure conditions, such as supercritical fluids and pressurized liquid extraction, offers several advantages for the extraction of bioactive molecules. These include the reduced use of toxic solvents, reduced extraction time, high selectivity, and the possibility of being applied in combination in a cascade of progressive extractions. In this review, an overview of high-pressure extraction techniques related to the recovery of high added value compounds from waste generated in food industries is presented and a critical discussion of the advantages and disadvantages of each process is reported. Furthermore, the possibility of combined multi-stage extractions, as well as economic and environmental aspects, are discussed in order to provide a complete overview of the topic.

**Keywords:** solvent power; supercritical fluids; bioactives

## 1. Introduction

The agri-food sector generates huge quantities of solid and liquid waste, containing a high organic load. Landfilling or disposal of such waste in an uncontrolled way can cause environmental pollution and economic loss. On the other hand, these residues containing valuable metabolites have an enormous potential for the production of sustainable additives, chemicals, biofuels, and energy [1]. In particular, high added value compounds that are still present in by-products of the food processing industry can be employed as food additives and for the formulation of new functional foods, nutra-/pharmaceuticals, cosmeceuticals, and beauty products, as well as for innovative and more sustainable packaging [2]. Efforts for the systematic reuse of food waste to produce commercial products and energy provided the integrated approach proposed by the biorefinery concept. Figure 1 reports a generic example of the biorefinery solution for food waste valorization.

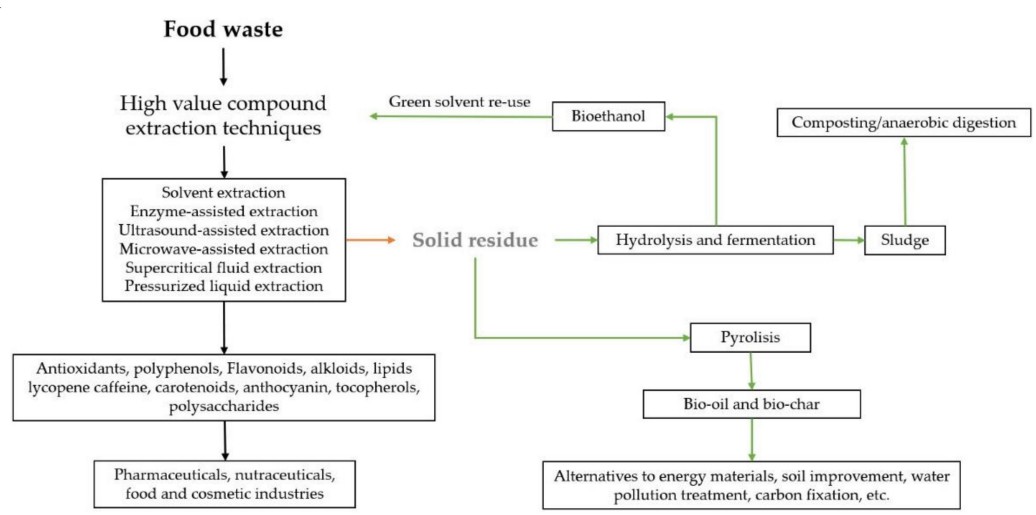

**Figure 1.** Biorefinery concept applied to food waste.

Agri-food industry by-products comprise several complex biomasses. The waste originated by various branches of the food industry can be divided into two main groups (plant or animal origin) and seven subcategories: cereals, root and tubers, oil crops and pulses, fruits and vegetables, meat products, fish and seafood, and dairy products [3,4]. In agreement with the hierarchical approach proposed by the biorefinery concept [5], these by-products contain valuable compounds that can be exploited in the development and production of new functional ingredients. In particular, interesting bioactive compounds present in food industry residues include proteins, polysaccharides, lipids, polyphenols, carotenoids, tocopherols, and essential oils.

Several extraction techniques have been proposed in the literature for their efficient recovery, from the conventional solid liquid extraction to non-conventional techniques, with the aim of improving the extraction yield of targeted compounds, avoiding deterioration and loss of functionality during processing, and allowing the food grade nature of the final product [6]. These non-conventional techniques include microwave-, ultrasound-, and supercritical fluid-assisted extraction, pressurized liquid extraction, pulsed electric field, and enzyme-assisted extraction techniques. In particular, the use of extraction performed under high-pressure conditions such as supercritical fluids and pressurized liquid extraction offers several advantages for the extraction of bioactive molecules, including short processing times, reduced use of organic solvent, and mild operative conditions able to avoid thermal degradation [7]. These technologies are considered environmentally friendly and are generally able to produce higher yields and high-quality final extracts.

Therefore, this review provides a comprehensive introduction to the literature update on pressure-assisted extraction processes. A critical analysis of the different processes proposed in the literature is reported, with the objective of defining the most suitable process for the extraction of different categories of high added value compounds that can be find in natural waste. Moreover, the possibility of combining different extraction processes in order to complete the valorization of agri-food waste is also analyzed for some case studies.

## 2. Pressurized Liquid Extraction Processes

In recent years, as a consequence of the deepening of research on biologically active natural substances, many extraction methods have been developed in order to obtain active compounds directly from natural sources. Phenolic compounds, carotenoids, flavonoids, lipids, proteins, polysaccharides, and essential oils are examples of active compounds obtainable from natural sources.

Among the emerging extraction technologies that can enhance the performance of conventional extraction techniques, pressured liquid extraction (PLE) is considered to

be environmentally friendly and efficient, and generates a small amount of waste while reducing cost and time [8,9]. The methodology is based on using solvents below their critical point in order to preserve the liquid phase during extraction. Firstly, PLE was introduced as Accelerated Solvent Extraction (ASE) technology by Dionex Corporation [8] in 1995 at the Pittcon Conference. It is also known as pressurized solvent extraction (PSE), enhanced solvent extraction (ESE) or pressurized fluid extraction (PFE) [10–13].

In the beginning, this technology was used for trace analysis of organic contaminants [8,14,15]. Currently, this technique is mainly applied to the extraction of environmental pollutants in soil, sediments, sewage sludge, and fly ash. However, in recent years, due to the advantages of PLE that include significantly shortened extraction time and low solvent volume required, easy operation, and good performance, this technology is being applied to different fields, including biology, pharmaceuticals, and the food industry [15–18]. As a result, the number of papers related to pressure-assisted processes in this field have increased significantly (Figure 2).

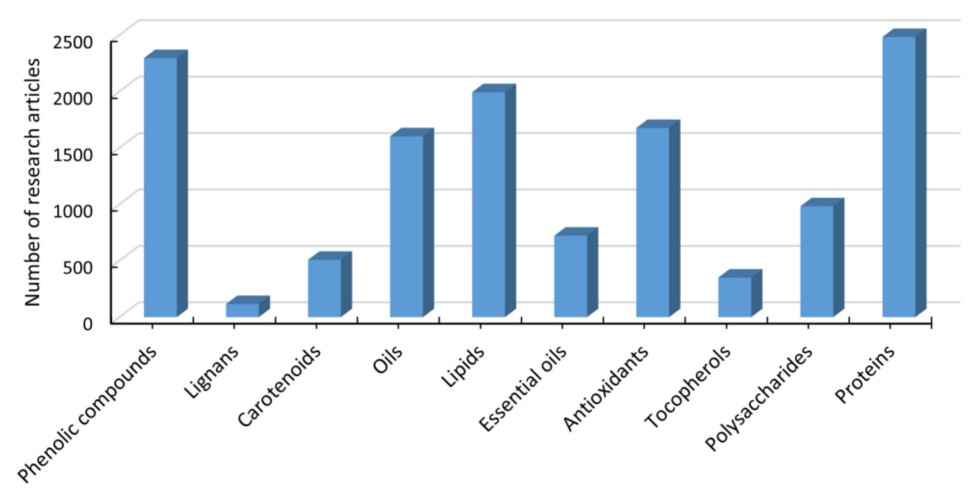

**Figure 2.** Number of research articles on the main active substances extracted by PLE from 2010 to 2021 (ScienceDirect).

This growing interest is mainly due to some impressive features of PLE; indeed, it is a process that can be automated, is characterized by reduced extraction time and solvent consumption, and its settings are particularly suitable for compounds that are sensitive to oxygen and light. PLE requires minimal sample pre-treatment, especially for non-fat samples such as homogenization and/or drying [7].

Regarding the mechanism of extraction, PLE combines elevated pressure and temperature with liquid solvents, without exceeding the critical points of the solvent or solvent mixtures, which remains in a liquid state. In this way, it is possible to increase the mass transfer rate from the matrix compounds to the extraction solvent to achieve a fast and efficient solid–liquid extraction. PLE often demonstrates better recovery when compared to other extraction techniques such as microwave-assisted extraction (MAE) or ultrasound-assisted extraction (UAE) [19,20]. Other significant advantages of PLE include the elimination of post-extraction steps, such as filtration and centrifugation, the high level of automation, and the possibility of coupling with different techniques, such as ultrasound [8,21,22].

Regarding the operation mode, there are three possible different modalities for applying the PLE process: static [23–25], dynamic [26], and static–dynamic. In the static mode, a fixed volume of extracting solvent is used; in the dynamic mode, the extracting solvent flows continuously through the sample; in the static–dynamic layout, a combination of the two modes is proposed. Figure 3a shows that static pressurized liquid extraction generally consists of the following components: (1) a pump to feed the solvent through the system; (2) an extraction vessel. The extractor consists of an extraction cell (EC) in which the solid matrix is contained, surrounded by a heating system. The layout is completed by

(3) a system cooling the fluid from oven temperature to room temperature or even lower; (4) an on/off pressure needle valve located after the cooler to maintain pressure during static extraction; (5) a $N_2$ flushing system, located between the high-pressure pump and the extraction system. The dynamic pressurized liquid extraction system is improved on the basis of the static pressurized extraction system, as shown in Figure 3b: (1) the selector valve connected to the $N_2$ flow is replaced by the inlet valve, because in dynamic mode, there is no need to purge the system; (2) the extraction vessel; (3) a restrictor connected to the outlet of the cooler (in place of the on/off valve) is used to maintain the constant pressure in the system [27].

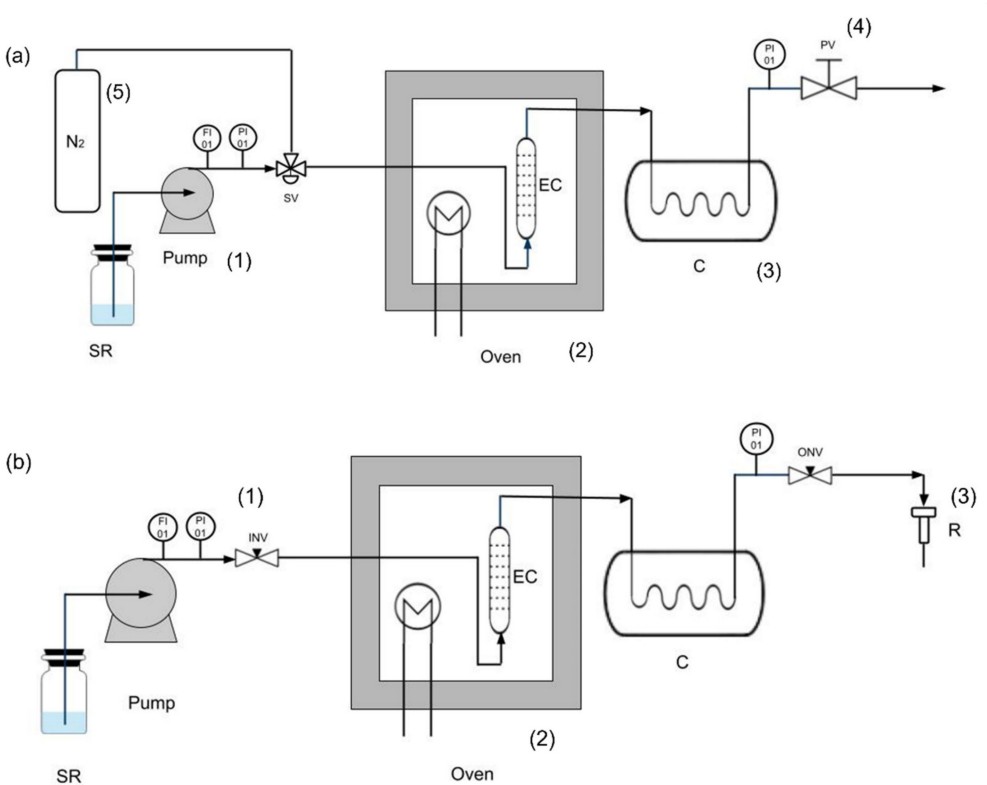

**Figure 3.** PLE system adapted from [27–29]. (**a**) Static mode; (**b**) Dynamic mode. SR, solvent reservoir; FI, flow indicator; PI, pressure indicator; SV, selecting valve; EC, extraction cell; C, cooler system; PV, pressure valve; INV, inlet needle valve; OV, outlet needle valve; R, restrictor.

Other configurations of the pressure-assisted process are also proposed in the literature.

Extractor Naviglio is based on the application of pressure (about 8–9 bar) and decompressions up to atmospheric values, creating a suction effect and a mechanical transport of target compounds from the solid to the solvent phase. The system is composed of two cylinders in which extraction is performed, with the cylinders communicating through a tube on which an electro-valve is intercepted. A pneumatic system working with compressed air moves two pistons to provide a pressure rise during a static stage, during which pressure is held constant for a while; this improves the penetration of the solvent within the solid and a dynamic stage, when the pistons are removed from the equilibrium position and alternately work generating negative pressure gradient between the inside and the outside of the solid matrix and promoting the dragging effect of the extraction [30].This system is designed to work at room or low temperatures, reducing the thermal stress for thermo-sensitive compounds [31].

Moreover, high-pressure and temperature-assisted extraction (HPTE), similar to PLE, exploits the effects of pressure to work at a high temperature for the mass transfer improvements, but a different setup of the extraction cell is employed. Indeed, extraction is carried

out in a stainless-steel vessel and a stirrer equipped by impellers provides an excellent and homogeneous mixing action [32]. The extractor can work up to 350 °C and 200 bar, with the two variables monitored by a thermocouple and a pressure gauge, respectively. The thermal sensor is connected to a temperature that maintains the variable at the desired set point value by an external electrical jacket. Before the extraction, an inert atmosphere is created inside the extraction vessel by bubbling inert gases for air removal with the aim of limiting degradation reactions of thermo-sensitive compounds. In HPTE, temperature and pressure are strictly related since the second factor corresponds to the vapor pressure of the solvent at the set point temperature [33].

Manosonication-assisted extraction is an example of pressure-assisted process coupled with other extraction aid, particularly the cavitation effects induced by ultrasounds. The system involves a double-jacket reaction chamber, a sensor, and PID controller for temperature control, while an ultrasonic generator and nitrogen gas controlled by a manometer was supplied to adjust the amount of external pressure applied to the reaction chamber through an inlet fitting [34].

Some authors use a combination of static and dynamic [11,35,36] layout to optimize the extraction process. Both modes have their own advantages and disadvantages. The most significant negative feature of dynamic pressurized extraction is that the solute in the extract is diluted, which requires a concentration step as post-processing. Whereas for the static extraction mode, the extraction efficiency depends on the solute mass transfer balance between the matrix and the extracting solvent [37]. The critical factors are the temperature and time of the extraction. Furthermore, in the static extraction mode, complete extraction might not be achieved due to the limited volume of the extraction fluid.

In general, homemade experimental apparatus is used for the extraction. However, ready-to-use equipment for PLE is available on the market (for example, Parr Instruments Company Illinois (USA), Fluid Management System Inc. Kentucky (USA)).

## 2.1. Effect of Operative Parameters

In order to improve the extraction efficiency of an active substance, the key parameters that can affect the efficiency of the extraction process must be investigated. For pressurized liquid extraction, the factors that affect the extraction include the type of residue, solvent type, extraction time, pressure, temperature, pH, liquid-to-solid ratio, stirring speed, particle size, and sample moisture [38,39].

Pressure-assisted extractions can employ a wide range of extraction temperatures, usually from room temperature to 200 °C, and pressures up to 200 bar [11]. Temperature is a very important parameter to improve the mass transfer rate of the target compound from the solid matrix to the extraction solvent. Due to the non-compressibility of the liquid (the change in density with pressure is usually insignificant), the effect of pressure does not have a direct effect on the solubility of the analyte in some cases. However, operating at higher pressures can ensure that the solvent remains in a subcritical liquid state, while pressure forces the solvent to penetrate into the matrix pores and facilitates the desorption of the target compound due to a synergistic effect with temperature. The combined effect of high temperature and high pressure provides conditions that are able to significantly increase the solubility and diffusivity of the solute, to perform the rupture of the solute-matrix bond, and to reduce the surface tension and viscosity of the solvent [40–43].

Solvent selection is a key step in the design of industrial extraction processes. The extraction solvent must be able to dissolve the target solute and minimize the co-extraction of other matrix components. In addition, compatibility with subsequent processing steps (extract purification, target molecule preconcentration, or analytical techniques) and solvent volatility (if extract concentration is required) must also be considered [9,44].

The use of green solvents can have a positive impact on the environmental sustainability of processes; however, extraction yield can be negatively impacted by the selection of solvent less performant compared to the classical solvent. However, when non-conventional processes are performed, operating conditions are able to provide extraction yields and

selectivity even higher than conventional processes using conventional solvents. Among the green solvents investigated in the literature, especially for agri-food waste recovery, there are subcritical water and pressurized ethanol aqueous solutions. Since they do not produce residues during the extraction process, they are non-toxic and eco-friendly.

Indeed, as reported by Aliakbarian et al. [45], subcritical water extraction ($100 \leq T \leq 140$ °C and $8 \leq P \leq 15$ MPa, for 130 min) provided a total flavonoid and polyphenol 12- and 19-fold higher than conventional solid liquid extraction performed with water at 25 °C for 19 h, and with a higher antiradical power. In addition, polyphenol extraction from *Teucrium montanum* L. carried out by subcritical water gave higher concentrations of total phenolics in comparison to aqueous, methanolic, acetone, ethyl acetate, and petroleum ether extracts obtained by conventional solid/liquid extraction [46]. At the same time, the comparison of HPTE with other extraction techniques, using ethanol as solvent, showed the best performance of the former on the recovery of bioactive compounds from *Arthrospira platensis,* while solid–liquid extraction provided the worst performance [47].

### 2.2. PLE of Bioactive Compounds from Waste

Food industry by-products and waste have been proven to be a cheap raw material for the recovery of compounds with commercial value. For example, molecules of interest for the industry are caffeine from spent coffee grounds [33,48], lycopene and carotenoids from tomato waste [49], astaxanthin from crustacean and shrimp waste [50,51], polyphenols from fruit waste and rice [52], and lipids and protein from the waste of olive or microalgae [53–55]. These active substances have potential health benefits and find applications as ingredients or additives in the medical field, as well as in the food and cosmetic industries. By enabling the use of green and food-grade solvents, pressure-assisted extraction has been successfully applied to extract active substances (Table 1) such as proteins, polysaccharides (such as lignans), lipids, polyphenolic compounds (carotenoids and tocopherols), and essential oils from food waste [18].

**Table 1.** Example for extraction of biocompounds with pressurized extraction. EtOH = ethanol; PWE = Pressurized water extraction; PHWE = Pressurized hot water extraction; SWE = Subcritical water extraction; AS = Agitation speed; S/L = solid to liquid ratio; HPTE = high- pressure high temperature extraction.

| Compounds | Source | Extraction Solvent | Operating Conditions | Sample Weight Treated | Extraction Method | Ref |
|---|---|---|---|---|---|---|
| Proteins | *Sambucus nigra* L. branches | water | 50 °C and 15 MPa for 5 min | 4 g | PHWE | [56] |
| Proteins | Wheat bran | water | 80 °C, pH 9.3, and 30 bar for 30 min | 15 g | SWE | [57] |
| Proteins | Spirulina | water | 40 C, pH 4, and 103.4 bar for 10 min | 500 mg | PLE | [58] |
| Proteins | Sea Bream (*Sparus aurata, Salmo salar* and *Dicentrarchus labrax*) Side Streams | water | 1500 psi; 20 °C, pH 7, 5 min for muscle; 60 °C, pH 4, 15 min for heads; 50 °C, pH 7, 15 min for viscera; 55 °C, pH 7, 5 min for skin; 60 °C, pH 7, 15 min for tailfins | 6 g | PLE | [59–61] |
| Proteins | Pomegranate peel (*Punica granatum* L.) | 70% (*v/v*) EtOH | 120 °C, 1500 psi static extraction time, 3 min; extraction time, 12 min | 10 g | PLE | [62] |
| Proteins | *Saccharina japonica* | water (1.13–2.40%) | 140 °C and 50 bar for 5 min | 10 g | PLE | [63] |

**Table 1.** *Cont.*

| Compounds | Source | Extraction Solvent | Operating Conditions | Sample Weight Treated | Extraction Method | Ref |
|---|---|---|---|---|---|---|
| Fucoidan | *Saccharina japonica* | NaOH (4.15%) | 140 °C and 50 bar for 5 min | 2 g | PLE | [63] |
| *L. barbarum* polysaccharides | Fruit of *Lycium barbarum* | water | 100 °C and 1500 psi for 20 min | 250 mg | PLE | [64,65] |
| Fucoidan | Saccharina japonica | water | 127.01 °C, 80 bar, S/L ratio of 0.04 g mL$^{-1}$, AS of 300 rpm, for 11.98 min | <20 g | SWE | [66] |
| β-glucan | *Tuber melanosporum* | water and 100% EtOH | 180 °C and 16.7 MPa for 30 min | 500 mg | PLE | [67] |
| Polysaccharides | *Fucus virsoides* and *Cystoseira barbata* | water and 0.1 M H$_2$SO$_4$ | 140 °C and 1500 psi for 15 min | 1 g | PLE | [38] |
| Polysaccharides | Sagittaria sagittifolia L. | water | 170 °C, L/S of 7 mL/g for 16 min | 1 g | SWE | [68] |
| Polysaccharides | Lentinus edodes | water | 150 °C for 15 min | 1 g | SWE | [69] |
| Gallic acid | Cardoon leaf | 100% EtOH | room temperature, 9 bar, static phase 2 min; dynamic phase 2 min | 40 g | Naviglio® method | [70] |
| Gallic acid | Cowpea | 50% (*v/v*) EtOH | 170 °C, 10.34 MPa for 10 min | 5 g | PLE | [71] |
| Gallic acid | *Fucus vesiculosus* | 58.65% (*v/v*) EtOH | 137.18 °C for 4.68 min | 1 g | PLE | [72] |
| Gallic acid | Grape marc of Croatina cultivar and olive pomace | 75% (*v/v*) EtOH | 180 °C for 90 min | 5 g | HPTE | [73] |
| Gallic acid | Spent coffee (*Coffea arabica* L.) | water | 110–190 °C; time: 15–75 min; solid-to-liquid ratio: 1:10–1:70, *w/v* | 4 g | SWE | [74] |
| Gallic acid | Grape skins | methanol | 150 °C for 270 min | <10 g | HPTE | [75] |
| Gallic acid | Biquinho pepper | 75% wt ethanol | inlet and outlet temperatures of 190 °C and 88 °C, respectively, 10 MPa | <5 kg | PLE | [76] |
| Gallic acid | Red wine pomace | 37.5%: 37.5%: 25% water: EtOH: CO$_2$ | 80 °C, 25 solvent to DW ratio (kg/kg DW), 100 bar for 30 min | <5 kg | PLE | [77] |
| Polyphenolic compounds | *Solanum stenotomun* Peel | 80% (*v/v*) of ethanol in water acidified to pH 2.6 with acetic acid | 65 °C, 100 bar | 10 g/min | PLE | [78] |
| Polyphenolic compounds | *Saccharina japonica* | NaOH (1.54–3.64%) | 140 °C and 50 bar for 5 min | 10 g | PLE | [63] |
| Polyphenolic compounds | Corn Silage | 10% (*v/v*) EtOH | 180 °C, L/S ratio 20 for 120 min | <10 g | HPTE | [79] |

**Table 1.** *Cont.*

| Compounds | Source | Extraction Solvent | Operating Conditions | Sample Weight Treated | Extraction Method | Ref |
|---|---|---|---|---|---|---|
| Caffeic acid | Spent coffee (*Coffea arabica* L.) | 50% (*v/v*) EtOH | 150 °C | 4 g | HPTE | [80] |
| Caffeic acid | Spent coffee (*Coffea canephora*) | 54% (*v/v*) EtOH | L/S ratio of 10 mL/g, 150 °C for 60 min | <10 g | HPTE | [48] |
| Polyphenolic compounds | Lotus seedpod | water | 140 °C, L/S ratio of 70 mL/g for 20 min | <10 g | SWE | [81] |
| Polyphenolic compounds | *Agave americana* leaves | water | 150 °C and 10 bar for 240 min | <10 g | HPTE | [82] |
| Polyphenolic compounds | Grape skins and defatted grape seeds | water | 80–120 °C, 10 MPa, 2–5 mL/min, for 2 h | 65 g | SWE | [83] |
| Polyphenolic compounds | Silybum marianum | EtOH | over half an hour | <10 g | Navi-glio® method | [84] |
| Polyphenolic compounds | Pomegranate peels | water | 40 °C and 102.1 atm for 5 min | 10 g | PWE | [85] |

### 2.2.1. Proteins

The growing world population requires a more sustainable source of protein for food and technological aims. Thus, new sources include insects, fungi, and algae, as well as waste streams from food processing [86]. Conventional methods for the recovery of proteins involve the use of alkali, since the basic environment allows the breakage of disulfide bonds in protein and higher solubility providing high extraction yields, increases their bioavailability and digestibility, but degrades protein quality [87]. In recent years, pressure-assisted extraction has been successfully applied to extract proteins from a variety of food processing residues. Šalplachta et al. [56] optimized pressurized hot water extraction conditions (50 °C, 15 MPa, and 5 min as extraction time) for the recovery of proteins from *Sambucus nigra* L. branches, demonstrating the highest yields and better reproducibility of the pressure-assisted technique compared to traditional solid–liquid extraction. Zhou et al. [58] used PLE to extract proteins and bioactive compounds from spirulina. Response surface methodology (RSM)-centered composite design (CCD) was used to evaluate and optimize the extraction time (5–15 min), temperature (20–60 °C), and pH (4–10) of the PLE process for extraction (103.4 bar). The results showed that under optimal extraction conditions (10 min, 40 °C, and pH 4), PLE significantly increased the protein yield of the extracts (46.8 ± 3.1%) compared to non-pressurized extraction.

PLE was also applied to produce extracts with antioxidant activity from *Sparus aurata* side streams. Protein recoveries of 22% (muscle), 33% (head), 78% (viscera), 24% (skin), and 26% (tailfin) were obtained, with values 1.2–4.5-fold higher compared to samples obtained from conventional stirred extraction [59]. Similarly, Fuente, et, al [60,61] used the PLE technique to obtain protein extracts with antioxidant capacity from muscle remains, head, viscera, skin, and caudal fins of salmonids. Protein recovery was ≈92% for all samples except viscera, implying a 1.5–4.8-fold increase compared to stirred extraction (≈28%). The effect of pressure in enhancing extraction yields of proteins can be ascribed to the alteration of covalent bonds and to the breakdown of proteins into soluble peptides, as well as to the induced variation of dielectric and physical properties of solvent (such as subcritical water) through the operating conditions enabled by high-pressure-assisted processes [88].

### 2.2.2. Polysaccharides

Polysaccharides are polymers composed of more than 10 monosaccharides with straight or branched glycosidic bonds [89]. They are characterized by low toxicity, low cost,

and high biocompatibility and biodegradability [90]. Natural polysaccharides have several potential applications in food industries due to their properties as stabilizers, emulsifying, thickeners, gelling agents, and film forming [64,90–94]. In addition, they exert bioactivities that can be exploited in medicine, including antioxidant, antitumor, anti-hyperglycemic, and immune regulation activities [95].

Plant polysaccharides' extraction efficiency is often reduced by the presence of cell wall polymers, so extraction aids able to disrupt the cell structure have to be applied to enhance extraction yields. The application of advanced extraction techniques such as PLE shows the great potential of polysaccharides recovery. The better efficiency towards polysaccharides was reported by Dobrinčić et al. [38], who evaluated the effect of MAE and PLE parameters (solvent, temperature, time, and number of cycles) on the polysaccharide yield and chemical composition (total sugars) of *Fucus virsoides* and *Cystoseira barbata*. The best PLE parameter was 0.1 M $H_2SO_4$ as solvent at 140 °C for two cycles of 15 min. In addition, PLE obtained a significant increase in polysaccharide yield from *Fucus virsoides* and the polysaccharides obtained using PLE had the highest polydispersity index, fucose, and sulfate content, and the lowest glyoxylate content; however, they had lower antioxidant activity. Herbst et al. [40] studied the yield of extracting valuable compounds from brewer's grains using PLE under constant pressure with varying temperature, solvent type, and flow rate, and the total phenolic compounds, total flavonoids compounds, antioxidant activity, reducing sugars, and total reducing sugars as indicators, and compared the results with the Soxhlet extract. The results show that PLE can maintain a high extraction rate of reducing sugars and total reducing sugars in the BSG matrix. The application of high pressure can modify the conformation and structure of polysaccharides, changing their biological activities [96,97]. High-pressure extraction was also coupled with ultrasounds, creating hybrid techniques such as high-pressure ultrasonic-assisted extraction [98], which was employed for the recovery of polysaccharides from *Hovenia dulcis*.

### 2.2.3. Lipids

Lipids are categorized as either neutral or complex. Neutral lipids include glycerides, wax esters, free fatty acids, free fatty alcohols, carotenoids, sterols, tocopherol and lignin-type antioxidants, and biologically derived hydrocarbons. Complex lipids include phospholipids, glycolipids, ceramides, cerebrosides, and gangliosides.

Many well-known extraction methods have been applied for the extraction of lipids from food waste, including Soxhlet, Bligh and Dyer, and Folch processes, and SFE; more recently, PLE have been successfully developed to enhance lipid extraction. Many applications of PLE methods have been reported, such as the extraction of compounds from plants [28,99–101], animals [102–105], soil [106,107], and algae [108], but few from yeast [109,110]. According to the large potential of lipid production with oleaginous microorganisms, the application of PLE methods to lipid extraction from yeast is a major challenge but with promising perspectives [111,112].

Yongjin et al. [113] used a PLE process to extract microalgal lipids from Isochrysis biomass. Compared with the Soxhlet and Folch methods and PLE using n-hexane, PLE using ethanol exhibits excellent lipid extraction performance. Cescut et al. [114] used PLE to extract total lipids from oleaginous yeast (*Rhodotorula glutinis*) and compared it with two traditional methods: the Soxhlet method and the improvement of the Bligh and Dyer method. The results showed that the extraction time of PLE was reduced by 10 times, the amount of solvent was reduced by 70%, and the lipid extraction rate was significantly higher than the reference method (Soxhlet method and Bligh and Dyer method). Villanueva-Bermejo et al. [115] used PLE to extract 3-rich oil from chia seeds by using pressurized food grade ethanol at 60 °C. In a short extraction time (10 min), the recovery rate of the seeds was close to 100% [106,116–121]. However, since lipids are soluble in non-polar solvents such as hexane, extraction using PLE cannot be performed applying green solvents. For this class of compounds, supercritical fluids extraction is more advantageous.

### 2.2.4. Polyphenolic Compounds

Among the secondary metabolites produced by plants, phenolic compounds are a relatively abundant category. Generally, phenolic compounds are most often classified into flavonoids and non-flavonoids. Flavonoids are the most abundant plant polyphenols consumed by humans. They are polyphenol compounds with two aromatic rings connected by a three-carbon bridge. For example, more than 800 phenolic compounds have been identified in plants. Chemically, phenolic compounds have at least one aromatic ring and one or more hydroxyl groups [122]. Although phenolic compounds are not necessary for human metabolism, if they are present in the diet, they can improve health and may reduce the risk of serious diseases. In fact, phenolic compounds are widely studied because they can reduce the reactive oxygen species that cause oxidative stress and are related to the prevention and treatment of cardiovascular and chronic degenerative diseases. In addition, phenolic compounds are due to their antibacterial, anti-inflammatory, antidiabetic, and anticancer properties [123–128]. At the same time, the antioxidant capacity of phenolic compounds indicates that they can be used as natural additives for functional foods. For example, the phenolic compounds recovered from olive mill wastewater [129] were evaluated as an agent to extend the shelf life of baked products due to their antibacterial properties. In the same way, Basanta et al. [130] extracted phenolic compounds from sugar-exhausted *Prunus avium* homogenate to be loaded into low-methoxyl-pectin film as an antioxidant barrier for food preservation.

However, the extraction of these compounds is challenging because they may be unstable and their biological activity can be affected by extraction process parameters and external factors such as the presence of oxygen and light. Phenolic compounds have been extracted for decades using traditional methods such as Soxhlet extraction and maceration [131–133].

Alternative and more effective extraction processes are needed to overcome these shortcomings and produce higher extraction yields while maintaining the integrity of the compound.

Due to the many advantages of pressure-assisted extraction processes, the research on the extraction of polyphenolic compounds by this technology has become increasingly intense in recent years.

Battistella Lasta et al. [134] recovered phenolic compounds from beetroot (*Beta vulgaris* L.) residues, leaves, and stems by PLE at a temperature of 40 °C, pressures of 7.5, 10 and 12.5 MPa, and a flow rate of 3 mL min$^{-1}$. Okiyama et al. [135] demonstrated that PLE was effective in enhancing the extraction rate of flavanols from industrial by-product cocoa shells. Shang et al. [136] optimized extraction conditions for total phenolics and caffeine from waste coffee grounds using the PLE method with water and ethanol.

Considering the thermosensitivity of several polyphenols, the positive effect of pressure in improving the extraction process is of a great interest, allowing good extraction yields even at low temperatures [137] or enabling the use of a very high temperature but for short times [138].

## 3. Supercritical Fluid Extraction

Supercritical fluid extraction (SFE) is an emerging clean technique that developed in the late 1970s/early 1980s and represents the most studied supercritical carbon dioxide-based technique [139]. Most of the published works focus on the extraction of compounds of interest from plant matrices such as seeds, fruits, or leaves. These studies have mainly been the subject of previous reviews. They are outside the scope of the present review paper, which instead focuses on the extraction of active compounds from food industry waste. The SFE process exploits the advantages of the use of carbon dioxide in the supercritical state: mild critical values of temperature and pressure (Tc = 31.1 °C; Pc = 73.8 bar); total elimination of the use of organic solvents that could remain present in traces in the extract (with a consequent limitation of the environmental impact of the extraction process); the possibility of simply modulating the solvent power of $CO_2$ which changes significantly

with pressure and temperature; ease in the scale-up of the process up to the industrial scale [140–144]. It is important to highlight that carbon dioxide in the supercritical state is a "hybrid" between a gas and a liquid; indeed, it has some properties typical of gases (such as high diffusivity, which guarantees a better transport of matter with consequent extraction times drastically reduced than the ones of the classic solvent extraction) and others comparable to those of liquids (such as density, to which the solvent power is directly connected). Moreover, carbon dioxide is non-toxic, non-flammable, available in large quantities, and cheap. The main disadvantage of supercritical carbon dioxide (scCO$_2$)-based techniques is linked to the investment costs (for the design of equipment that operates at high pressures) and operational costs (for pumping CO$_2$). The fundamental elements in an extraction plant are sketched in Figure 4: (a) the carbon dioxide vessel; (b) the pump (with refrigerated head to avoid cavitation phenomena); (c) a heat exchanger to preheat the carbon dioxide; (d) an extraction vessel in which the matrix from which to extract the compounds of interest is loaded; (e) one or more separators at lower pressures than the extractor's one.

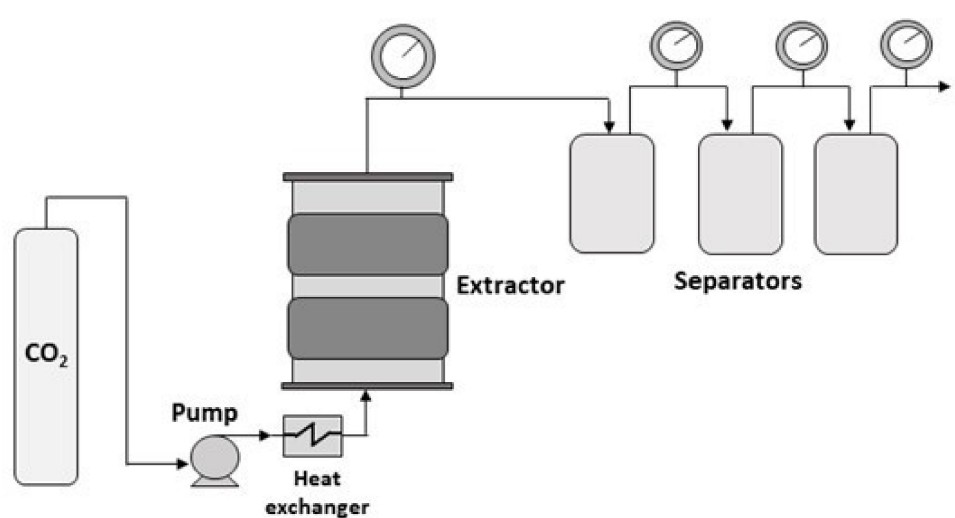

**Figure 4.** Sketch of the SFE plant.

In some cases, due to the low polarity of carbon dioxide, a polar cosolvent can be added to increase the solvent power of the mixture. With this strategy, however, one of the advantages of SFE, which is the complete elimination of organic solvents, is lost.

In general, homemade experimental apparatus is used for the extraction. However, ready-to-use equipment for PLE is available on the market (for example, Extratex and SFE process).

### 3.1. Effect of the Operating Parameters

The effect of pressure, temperature, average particle size of the matrix, supercritical extraction time, amount of cosolvent, and CO$_2$ flowrate on the extraction yield (g active compounds obtained/g raw material × 100%) is generally considered [17]. Typical values of the operating conditions for the SFE process are pressures in the range 300–500 bar and temperatures in the range 40–80 °C.

### 3.2. SFE of Bioactive Compounds from Waste

SFE has been widely used recently to reduce the environmental impact of agro-industrial waste. Residues, waste, and by-products generated by the food-growing and -processing industries have proven to be a valuable source of biologically active compounds that can be used in the food, nutraceutical, cosmetic, and pharmaceutical industries. Among the most widespread, there are certainly the waste of the tomato [145], fishery [146], and wine [147] industries; however, an important source of active compounds are also crops

and waste from less widespread productions such as cherimoya [148], peel and leaves, or defatted Assaí berries [149].

Industrial tomato by-products, for example, are rich in highly valued compounds such as carotenoids (lycopene, β-carotene), tocopherols, fatty acids, and sitosterols which can be selectively extracted simply modulating the operating conditions. Vági et al. demonstrated that if industrial tomato waste are processed with $scCO_2$ at 460 bar and 80 °C, the extract contained the highest concentration of carotenoids with 90.1% of lycopene, while products rich in tocopherols and phytosterol were obtained above 30 MPa and 40 °C [150].

SFE in some cases has been coupled with other processes to obtain value-added products from food waste. Yu et al. [151], for example, valorized corn, taro, lettuce, and bean sprout using a sequential SFE and thermochemical conversion. The obtained extracts, depending on the vegetal source, were rich in sterols with potential anticancer properties, difatty acids for acne treatment, or unsaturated fatty acids for cholesterol control.

3.2.1. Lipids

Lipids have typically been extracted from fruit, plant, coffee, or seafood processing residues using $scCO_2$ without the need for cosolvents; indeed, lipids are well soluble in $scCO_2$. Fatty acids (FA), common components of complex lipids, can be extracted from red fruits' seeds and peels or from citrus plants' seeds. In some papers, the results obtained through supercritical extraction and extraction using organic solvents, such as hexane, have been compared [152,153].

It is well known that red fruits such as raspberry, blueberry, wild strawberry, pomegranate, blackberry, and blackcurrant are very rich in antioxidants, in particular, flavonoids, polyphenols, carotenoids, anthocyanins, and vitamins C and E, which help counteract damage from free radicals, oxidative stress, and cellular degeneration. Waste from the industries that process these fruits (seeds and peels, for example) are rich in FA. Although the total yields in the extract are in some cases higher using hexane as the extracting solvent, the affinity of supercritical $CO_2$ for lipophilic compounds leads to purer extracts, with a higher percentage of fatty acids compared to hexane. For example, Campalani et al. demonstrated that when processing wild strawberry waste, 26.0% of fatty acids were obtained with $scCO_2$ and 1.4% using hexane [152].

Citrus (e.g., mandarin, lemon, lime, orange, tangerine, bergamot, and grapefruit) seeds are commonly considered as agro-industrial waste, but they are a source of by-products given they contain phenolic compounds, limonoids, carotenoids, and tocopherols [154,155]. Rosa et al. [153] compared the extraction of fatty acid and triacylglycerol content by processing mandarin, lemon, and grapefruit seeds with $scCO_2$ extraction and Soxhlet extraction using hexane. Compared to what was observed for the extraction from seeds and peels of red fruits, and also citrus seeds, the total yield was higher for Soxhlet than SFE; the fatty acid profiles obtained through the two extraction techniques were similar but, using SFE, no traces of unwanted solvent were present.

SFE has also been used to extract fatty acids from less common fruit-processing waste worldwide, such as tamarillo (native fruit from Peru, Ecuador, and Colombia), date (cultivated and consumed in African countries), rosehip, and mango. The processing residue from the preparation of tamarillo juices consists of the pericarp and seeds, which represents 24% of the fruit; rosehip seeds constitute about 30% wt of the fruit and is a waste product in the industrial production of herbal tea, fruit juice, jelly, jam, and marmalade; in the case of mango, 35–40% of waste is generated in the form of peels, kernels, and culled fruits [156–158].

Depending on the structure of the fruit waste, different SFE optimal operating conditions were identified. For tamarillo, the best results for the extraction of linoleic, oleic, and palmitic acids were obtained at 380 bar and 64 °C [156]; for saturated, monounsaturated, and polyunsaturated fatty acids (with oleic acid as the major constituent) extracted from date stones oil, the optimal conditions were 250 bar and 60 °C [159]; fatty acids were

successfully extracted from rosehip seeds at 300 bar and 40 °C [157]; for mango kernel, the highest yield was achieved at a higher pressure (500 bar) and a temperature of 40 °C [160].

In relation to the valorization of agro- and food industrial waste, some research groups are trying to valorize coffee by-products, since coffee is one of the world's most widely consumed beverages and its processing chain leaves behind large volumes of waste [161]. In this framework, Ahangari and Sargolzaei extracted fatty acids from spent coffee grounds through supercritical fluid, Soxhlet, microwave irradiation, and ultrasonic irradiation extraction [162]. By using SFE, they observed that higher yields were obtained at a higher pressure, lower temperature, and using hexane as a modifier (rather than water or ethanol). All the yields obtained using SFE were higher than those of the other extraction methods using organic solvents except the Soxhlet extraction. Considering the fatty acid compositions, the authors observed that linoleic, palmitic, oleic, and stearic acids were among the most present; in the case of SFE, different operating conditions led to significant differences among several fatty acids, whereas no significant differences were observed in the FA compositions of the extracted oils using organic solvents. FA, above all linoleic and palmitic acids, were extracted from coffee silverskin (the thin layer directly in contact with the coffee bean, which is the only residue from the operation of roasting) by Nasti et al. [163], using both SFE and Soxhlet extraction. From these authors' findings, it is possible to observe that yields are similar considering the conventional extraction and SFE at the best operating conditions; however, it should be highlighted that when using SFE, the extraction time was drastically reduced and the use of hydrocarbon or chlorinated solvents was avoided.

Undoubtedly, among the food waste containing fatty acids, the fishing industry also has to be considered. In fact, during processing, up to 50% of the body weight of the fish may remain as waste. The oil content of fish viscera is an important source of polyunsaturated fatty acids that have been recovered through SFE from the viscera of Indian mackerel [164] (*Rastrelliger kanagurta*) or of yellowtail fish [165] (*Seriola quinqueradiata*). When viscera were taken from Indian mackerel, the highest recovery was obtained using ethanol as a cosolvent and a pressure of 350 bar and temperature of 60 °C, with a yield close to the one obtained with Soxhlet extraction [164]. Considering the extraction from yellowtail fish, Soxhlet extraction with ethanol has a higher yield than SFE (56% vs. 41%), but omega-3 fatty acids extracted by SFE (at 300 bar and 40 °C) have a better oxidative quality than conventionally extracted oils [165]. Special consideration has to be made for carotenoids, since several applications of SFE can be found for the recovery of these natural bioactives.

### 3.2.2. Carotenoids

Carotenoids are a group of phytochemicals that are responsible for the yellow, orange, and red colors of foods and vegetables [166]. They can be classified in two broad subcategories: carotenes (such as α-carotene, β-carotene, and lycopene) and xanthophylls (such as lutein, astaxanthin, and cryptoxanthin). From the chemical point of view, xanthophylls are carotenes' oxygenated derivatives. Indeed, they differ for the presence of oxygen atoms in the structure: carotenes are purely hydrocarbons (for example, α-carotene, β-carotene, and lycopene formula is $C_{40}H_{56}$), whereas xanthophylls are formed by carbon, hydrogen, and oxygen (lutein formula is $C_{40}H_{56}O_2$, astaxanthin is $C_{40}H_{52}O_4$, and cryptoxanthin is $C_{40}H_{56}O$).

Although PLE has been used to extract carotenoids, the non-polar nature of some of these components makes SFE using $scCO_2$ the most suitable extraction technique for this category of compounds [17]. Supercritical carbon dioxide has been used alone or with low quantities of ethanol as cosolvent to extract those substances from different sources, such as tomato, vegetables, or seafood waste. Table 2 provides an overview of studies focused on the supercritical carbon dioxide extraction of carotenoids from food waste. The source from which the active principle has been extracted, the eventual amount of cosolvent added to $scCO_2$, the operating conditions in terms of pressure, temperature and extraction time, and the recovery of the carotenoid are reported. The recovery values reported in the table are

the ratios between the dry weight of the carotenoid in the extracts and the dry weight of the carotenoid contained in the original sample.

**Table 2.** Extraction of carotenoids by supercritical carbon dioxide. EtOH = ethanol; P = operating pressure; T = operating temperature; t = extraction time.

| Carotenoid | Source | Sample Weight Treated | EtOH % | P Bar | T °C | t min | Recovery % | Ref |
|---|---|---|---|---|---|---|---|---|
| α-carotene | Apricot peels | 5 g | 15.5 | 350 | 59 | 30 | 97.9 | [167] |
| β-carotene | Tomato paste waste | 60 g | 5 | 300 | 65 | 120 | 50 | [168] |
| β-carotene | Tomato processing waste | 40–50 g | / | 300 | 80 | / | 88 | [169] |
| β-carotene | Sweet potato peels | 5 g | 15.5 | 350 | 59 | 30 | 99.8 | [167] |
| β-carotene | Tomato peels | 5 g | 15.5 | 350 | 59 | 30 | 96.9 | [167] |
| β-carotene | Apricot peels | 5 g | 15.5 | 350 | 59 | 30 | 99 | [167] |
| β-carotene | Pumpkin peels | 5 g | 15.5 | 350 | 59 | 30 | 88.2 | [167] |
| β-carotene | Peach peels | 5 g | 15.5 | 350 | 59 | 30 | 99.2 | [167] |
| lycopene | Tomato paste waste | 60 g | 5 | 300 | 55 | 120 | 54 | [168] |
| lycopene | Tomato processing waste | 40–50 g | / | 300 | 80 | / | 80 | [169] |
| lycopene | Tomato processing waste | 2 kg | / | 450 | 70 | 420 | 75 | [170] |
| lycopene | Tomato peels | 5 g | 15.5 | 350 | 59 | 30 | 92.5 | [167] |
| lycopene | Apricot peels | 5 g | 15.5 | 350 | 59 | 30 | 83.2 | [167] |
| lutein | Spinach by-products | 25 g | 10 | 390 | 56 | 215 | 72 | [171] |
| lutein | Sweet potato peels | 5 g | 15.5 | 350 | 59 | 30 | 99.8 | [167] |
| lutein | Peach peels | 5 g | 15.5 | 350 | 59 | 30 | 75.3 | [167] |
| chlorophyll | Spinach by-products | 25 g | 10 | 390 | 56 | 215 | 50 | [171] |
| astaxanthin | Crustacean waste | 25 g | 13 | 200 | 40 | 180 | 62 | [50] |
| astaxanthin | Shrimp waste | 60 g | / | 250 | 45 | 120 | 47 | [51] |
| fucoxanthin | Brown seaweed waste | 56 g | / | 400 | 40 | 180 | 80 | [172] |

It can be noted that for the extraction of this class of compounds, in some cases $CO_2$ alone is used; in most cases, the addition of a cosolvent is required, which in all cases is ethanol. The amount of ethanol (when present) varies from 5 to 15.5%. Carotenes are extracted in general at 300–350 bar (only in one case has a pressure of 450 bar been adopted to extract lycopene [170]), whereas in the case of xanthophylls, a wider range of pressures can be observed (astaxanthyn is extracted from crustacean waste at 200 bar [50] and focoxanthin is extracted from brown seaweed waste at 400 bar) [172]. From the perspective of operating temperatures, carotenes are extracted at higher temperatures (from 55 to 80 °C) than are xanthophylls (where the range of 40–59 °C is sufficient). The recovery values are generally high, confirming that sc-$CO_2$ extraction is the most appropriate technique for carotenoids.

## 4. Combined Extractions

Pressurized Liquid Extraction (PLE) and Supercritical Fluid Extraction (SFE) are environmentally friendly advanced techniques enabling the use of non-toxic and/or GRAS solvents such as water, ethanol, or carbon dioxide for the extraction and enhancement of target compounds yield [173,174]. These green technologies are now highly encouraged due to recent increased attention to the environmental impact of processes.

The supercritical fluid extraction of polar compounds, in general, is conducted using an organic solvent as the modifier (frequently ethanol). For example, bioactive compounds are extracted from mandarin peel [175] using scCO$^{2+}$ 5 and 10% ethanol, from papaya [176] seeds using scCO$^{2+}$ 2–8% ethanol, from walnut husk [177] using scCO$^{2+}$ 20% ethanol, or from olive residues [178] (leaves, pomace and prunings) using scCO$^{2+}$ 60% ethanol. The results in terms of total phenolic compounds and antioxidant activity were, in some cases, compared to the ones obtained through conventional extraction: when phenolic antioxidants were extracted from papaya agro-industrial waste, the yield in the ethanolic extract was slightly higher than the one in the scCO$^{2+}$ ethanol extract, but the latter had

higher antioxidant activity in vegetable edible oil [176]. In the case of walnut green husk, SFE gave far more encouraging yields than those obtained using water, ethanol, or methanol as an extracting solvent [177]. Jafarian Asl et al. [179] compared the extraction of phytosterols and tocopherols from rapeseed oil waste by using $scCO^{2+}$ ethanol and conventional Soxhlet extraction. At the optimized operating conditions, which for SFE are a temperature of 40 °C, a pressure of 350 bar (for phytosterols) and 400 bar (for tocopherol), and 5 wt% ethanol as cosolvent, SFE resulted in efficiencies that were three times higher than the ones obtained with the modified Soxhlet extraction technique. Considering that a cosolvent is needed, the benefit of using $scCO_2$ is reduced. For this reason, PLE is preferred for the extraction of these compounds. Indeed, because sc-$CO_2$ preferentially extracts non-polar compounds, the residue from supercritical fluid extraction (SFE) can be used as feedstock for a second extraction stage based on PLE (or other pressurized-assisted extraction processes) to complete the extraction of polar molecules. Furthermore, the future direction of food waste valorization is directed towards the combination of these innovative processes in a sequential stage extraction [16,180]. This approach was evaluated by de Aguiar et al. [76], who, from an economical point of view, investigated the possibility of coupling PLE and SFE for the recovery of phytonutrients from biquinho pepper, using supercritical carbon dioxide and ethanol 75% aqueous solution. They estimated that producing biquinho pepper-derived products was economically viable and a combined and sequential process led to a lower manufacturing cost than the commercialization price of the extracts. Similarly, Viganò et al. [181] compared two process alternatives to obtain four extract fractions from passion fruit bagasse. The first, which showed economic promise, involved a sequential multi-stage process comprising three steps of supercritical fluid extraction (SFE) and one step of pressurized liquid extraction (PLE), while the second option included the production of one extract from passion fruit rinds by single-stage PLE process. The efficiency of the biorefining process, based on consecutive supercritical carbon dioxide and pressurized liquid extractions, was also proposed in the work of Tamkutė et al. [182]. The authors used cranberry pomace as a raw material for the recovery linolenic, linoleic acids, and tocopherol rich extract by SFE and ethanol-soluble and water-soluble antioxidants from a sequential PLE extraction using ethanol and water, respectively. The feasibility and efficiency of using a sequential process with green extraction methods based on GRAS solvents from the Amazonian munguba seed was also investigated, showing a free-solvent oil mainly composed of palmitic acid can be firstly recovered by SFE, followed by a phenolic-rich fraction obtained from the defatted cake using ethanol by PLE [101]. The same approach was evaluated for several biomasses [173,183,184], indicating the increasing research interest towards these integrated green-based processes. Nevertheless, different combinations of these high-pressure-assisted processes can be taken into account, to exploit the advantages of supercritical carbon dioxide-based processes to provide solvent-free extracts. This was proposed in the work of Villanueva Bermejo et al. [185], where extract obtained by PLE with ethyl lactate was further processed by supercritical carbon dioxide antisolvent technique for the solvent removal, or extract fractionation [140].

Although PLE and SFE are considered "green extraction technologies", a technology needs to be evaluated economically and be environmentally friendly before it can be industrialized. A tentative economic evaluation and comparison of supercritical fluid and pressurized liquid extraction analyzing single-stage and continuous-stage processes for phytonutrient extraction from red pepper is reported by Aguiar et al. [76]. It was observed that PLE is more economical for the same yield of target compound. Similarly, this result is consistent with the economic evaluation of the extraction of paclitaxel from passion fruit bagasse by Ferreira et al. [186]. The SFE process produces greater economical consumption. According to some research results, the PLE method is the most environmentally friendly, while the SFE method contributes the most to the environmental impact category. This is because in an extraction scenario, carbon dioxide needs to reach a high pressure to convert it into a supercritical fluid. The LCA results indicate that the equipment used to pump/compress and cool the $CO_2$ stream is responsible for these energy loads, making it a

decisive factor in the high environmental loads created by the SFE process. This technology can reach an economic convenience at a large scale of production (for example, industrial coffee decaffeination using SCF). To reduce energy consumption, the operation in a closed loop with total recirculation of $CO_2$ can be used in an isobaric condition in order to avoid the decompression, cooling, and recompression of $CO_2$ [187].

## 5. Conclusions

High-pressure-assisted extraction is a powerful tool for the extraction of different categories' compounds. In particular, high-pressure liquid extraction and supercritical fluid-assisted extraction can be complementary to obtaining the complete exploitation of a food waste material before disposal. These can lead to a double effect: on the one side, the possibility of reducing food waste and on the other side, the overall environmental impact reduction.

Thanks to the high operative pressures that characterize both the PLE and SFE processes, the main advantages of these innovative techniques compared to the traditional extraction processes are a high extraction yield, better selectivity, reduced use of toxic solvent (and reduced solvent residue in the final extract), and a better preservation of sensible compounds in a short time. However, the main limitations of the high pressure operation is that on the one hand it can improve the extraction but on the other hand it causes an increase in the overall plant cost, when compared to conventional extraction methods. In addition, costs related to the security assurance of the process have to be considered and this is also related to the high operative pressure. The variable costs can be considered as practically the same as for conventional extraction; in particular, in the SFE, the cost of $CO_2$ is not considered in the option of the closed-loop operation and in the PLE, the solvents can also be completely recovered after the extraction. The general increase in the investment cost related to the high-pressure operation can be balanced from the other side given that in these processes a product with high purity can be obtained, reducing the need for post-processing operations.

Beside the several mentioned advantages over conventional extraction processes, these technologies have mostly been applied only at a lab scale. For this reason, research should be directed towards proving the validity of these high-pressure-assisted processes in larger-scale production in order to promote their industrialization.

**Author Contributions:** Conceptualization, R.C. and I.D.M.; methodology, R.C. and I.D.M.; investigation, J.L.; resources, P.P.; data curation, J.L.; writing—review and editing, R.C., M.P. and I.D.M.; supervision, M.P.; project administration, P.P. All authors have read and agreed to the published version of the manuscript.

**Funding:** This research received no external funding.

**Institutional Review Board Statement:** Not applicable.

**Informed Consent Statement:** Not applicable.

**Data Availability Statement:** Not applicable.

**Conflicts of Interest:** The authors declare no conflict of interest.

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
