# Peer review of "High-Pressure Technologies for the Recovery of Bioactive Molecules from Agro-Industrial Waste"

_applsci, doi:10.3390/app12073642_

Round 1
Reviewer 1 Report
This paper, entitled High pressure technologies for the recovery of bioactive molecules from agro-industrial wastes, is a scholarly work and can increase knowledge on this domain. The authors provide an interesting review on the domain and the content is relevant to Applied Sciences. The abstract and keywords are meaningful. The manuscript is quite well written and well related to existing literature.
I have some specific and general comments:
- The authors should discuss about the several scales of work for the data used. It could be interesting to provide the information about the scale (labscale, pilot scale, industrial scale).
- Moreover it could interesting to provide the details about the amount of samples treated by each technology. Is it on grams, kilograms, tons? Please provide such details.
- The authors should provide also names and parameters of devices avalaible on market, with technical specifications and names of suppliers. It could improve the qualitiy of the manuscript, giving applicative consideration.
- Please discuss about CAPEX and OPEX, and provide costs analysis of such approach. Please discuss also gains vs limitations.
- Please discuss also about energy balance (energy consumption) and safety.
As it, the manuscript is not fully acceptable for publication and requires some amendments. I recommend the following decision: RECONSIDER AFTER MAJOR REVISION.
Author Response
Reviewer 1
This paper, entitled High pressure technologies for the recovery of bioactive molecules from agro-industrial wastes, is a scholarly work and can increase knowledge on this domain. The authors provide an interesting review on the domain and the content is relevant to Applied Sciences. The abstract and keywords are meaningful. The manuscript is quite well written and well related to existing literature.
We thank the reviewer for the positive evaluation of the review.
I have some specific and general comments:
- The authors should discuss about the several scales of work for the data used. It could be interesting to provide the information about the scale (labscale, pilot scale, industrial scale).
- Moreover it could interesting to provide the details about the amount of samples treated by each technology. Is it on grams, kilograms, tons? Please provide such details.
We thank the reviewer for the interesting suggestion. However, the majority of results reported and discussed in the review were obtained on lab scale plant. Only few applications of these processes up to pilot and industrial scale are reported in the literature. These examples of larger scale application are reported in Table 1 and Table 2.
- The authors should provide also names and parameters of devices avalaible on market, with technical specifications and names of suppliers. It could improve the qualitiy of the manuscript, giving applicative consideration.
The indications about the devices available on the market have been provided at the end of section 2 and at the end of section 3.
- Please discuss about CAPEX and OPEX, and provide costs analysis of such approach. Please discuss also gains vs limitations.
- Please discuss also about energy balance (energy consumption) and safety.
Following reviewer’s suggestion in section 4 and section 5 new paragraphs have been inserted in order to provide a useful comparison with conventional extraction techniques and discuss about advantages and principal limitations of these new technologies with particular consideration of economic, energetic and safety issues.

Reviewer 2 Report
Review presents a nice summary of high-pressure extraction technologies applied to agro-industrial wastes. The topic is of high relevance due to the important need of recycling/valorizing agro/food wastes and the constant development of new technologies. Review is of interest for the readership of the journal.
A more emphasized focus of the scope of the review was missing in certain part of the review: abstract, introduction, examples, conclusions. Chosen examples and applications are very interesting but discussion of advantages of high pressure emerging technologies compared with conventional is missing.
Conclusions are too general, they could be enriched with comments about future trends of high-pressure extraction technologies, e.g. kind of wastes, applications where improvement is needed, LCA and energy balances of high pressure technologies compared with conventional extraction technologies
Author Response
Reviewer 2
Review presents a nice summary of high-pressure extraction technologies applied to agro-industrial wastes. The topic is of high relevance due to the important need of recycling/valorizing agro/food wastes and the constant development of new technologies. Review is of interest for the readership of the journal.
A more emphasized focus of the scope of the review was missing in certain part of the review: abstract, introduction, examples, conclusions. Chosen examples and applications are very interesting but discussion of advantages of high pressure emerging technologies compared with conventional is missing.
Conclusions are too general, they could be enriched with comments about future trends of high-pressure extraction technologies, e.g. kind of wastes, applications where improvement is needed, LCA and energy balances of high pressure technologies compared with conventional extraction technologies
We thank the reviewer for the positive evaluation of the work. We modified the manuscript abstract and introduction trying to emphasize the focus of the work.
Section 4 and Section 5 have been enlarged for a comparison with conventional extraction techniques with a discuss about advantages and principal limitations of these new technologies with particular consideration of economic, energetic and safety issues.

Round 2
Reviewer 1 Report
The authors provide a revised version of their manuscript taking into account all the comments and requests of amendments made in the previous review. I agree with all the comments and answers given. As it, the manuscript is now fully acceptable for publication and I recommend the following decision: ACCEPT IN PRESENT FORM.